# Adaptive Training Distributions with Scalable Online Bilevel Optimization

## Abstract

Large neural networks pretrained on web-scale corpora are central to modern machine learning. In this paradigm, the distribution of the large, heterogeneous pretraining data rarely matches that of the application domain. This work considers modifying the pretraining distribution in the case where one has a small sample of data reflecting the targeted test conditions. We propose an algorithm motivated by a recent formulation of this setting as an online, bilevel optimization problem. With scalability in mind, our algorithm prioritizes computing gradients at training points which are likely to most improve the loss on the targeted distribution. Empirically, we show that in some cases this approach is beneficial over existing strategies from the domain adaptation literature but may not succeed in other cases. We propose a simple test to evaluate when our approach can be expected to work well and point towards further research to address current limitations.

## 1 Introduction

Large models pretrained on massive, heterogeneous datasets have impacted various application domains (Bommasani et al., 2021), including natural language processing (Devlin et al., 2019), computer vision (Mahajan et al., 2018), and audio processing (Schneider et al., 2019). These models are typically trained on two different distributions: a *generic distribution* for pretraining and a *specific distribution* for fine tuning. Only the specific distribution matches the test conditions while the generic distribution offers an abundant source of data with some similarities to the specific data. This novel paradigm builds upon earlier work in multitask learning (Caruana, 1997), transfer learning (Bennett et al., 2003), and domain adaptation (Moore & Lewis, 2010). For all of these methods, the accuracy of a model on the specific task heavily depends on selecting an appropriate distribution over the generic auxiliary tasks and data.

This work proposes a scalable online strategy for data selection along with a comprehensive and realistic empirical study. We build upon a bilevel formulation of the generic re-weighting problem which allows for gradient-based optimization (Franceschi et al., 2018).

Our contributions are several. First, we unify several gradient-based data selection methods into a common framework in which their similarities and distinctions are more easily understood. Second, we introduce a scalable, online algorithm. This algorithm can train a large model while updating an inexpensive auxiliary data selection model which tracks the distribution required to make fast progress on the targeted task. Our algorithm leverages the asymmetry in computational cost between the selection model and the large model by filtering examples on the fly, ensuring that the majority of examples are not examined by the large model.

Third, we perform a comprehensive and realistic empirical comparison of data selection strategies. We compare several alternative strategies across different tasks and modalities including large scale language modeling, machine translation, and image classification. Finally, we propose a simple metric based on gradient alignment that correlates with the success and failure of gradient-based data selection methods.

## 2   Related Work

Prior work has proposed automatic methods to adjust the generic training distribution in order to improve model generalization on the specific task. The domain adaptation literature has explored variants of importance sampling, which uses importance weights to emphasize or select some generic examples. These weights have been determined via domain classifiers (Aharoni & Goldberg, 2020; Gururangan et al., 2020), via gradient alignment and fine-tuning (Wang et al., 2018; Grangier & Iter, 2022), or via the estimation of the label distribution (Ngiam et al., 2018). Related to domain adaptation, the removal of label noise in the generic distribution has received attention with methods based on influence functions (Koh & Liang, 2017; Pruthi et al., 2020; Schioppa et al., 2022), data models (Ilyas et al., 2022; Jain et al., 2022), and data Shapley values (Ghorbani & Zou, 2019; Karlaš et al., 2022).

As an alternative to static weighting, the literature also explored dynamic weighting where the distribution over generic examples is adapted during training. The two primary strategies are reinforcement learning and direct optimization. Reinforcement learning does not assume that the specific task loss can be differentiated with respect to the weighting parameters. Instead, a parameterized model of the generic distribution is adjusted through reinforcement learning: the current model proposes generic distributions, and their reward is measured as the specific loss after a few steps of generic training over a proposal distribution (Kumar et al., 2019; Yoon et al., 2020; Zhu et al., 2020). On the other hand, direct optimization assumes a differentiable functional dependency between the weighting parameters and the specific training loss. This dependency can be derived through meta learning by unfolding the generic update (Ren et al., 2018; Hu et al., 2019; Shu et al., 2019; Zhang & Pfister, 2021): one gradient update step minimizing the weighted generic loss depends on the weighting parameters. The impact of this update can be evaluated by computing the post-update specific loss which can then be differentiated with respect to the weighting parameters. As an alternative to update unfolding, a bilevel formulation of the reweighting problem also allows for direct optimization (Franceschi et al., 2018). Our work builds upon this bilevel formulation.

Other research areas intersect with generic sample reweighting. Prior work considered learning a distribution over training data augmentations (Ho et al., 2019; Lim et al., 2019; Zoph et al., 2020). Curriculum learning has also been introduced to visit successive training distributions based on training instance difficulty (Bengio et al., 2009; Kumar et al., 2010; Jiang et al., 2018; Saxena et al., 2019). Multi-task learning research has considered gradient projection to minimize negative interactions between tasks (Yu et al., 2020; Dery et al., 2020; Liu et al., 2021). Importance sampling for accelerated stochastic training (Zhao & Zhang, 2015; Katharopoulos & Fleuret, 2018) is also relevant.

## 3   Problem Setting

Classical machine learning assumes that the model is trained on data drawn from the distribution from which the test data will also be sampled from (Vapnik, 1999). Our setting is different and belongs to the field of transfer learning (Caruana, 1993; Thrun & Pratt, 1998). In our setting we are given two training sets, a large generic training set $\mathcal{D}_{\text{generic}}$ and small specific training set $\mathcal{D}_{\text{specific}}$. Only the latter set is representative of the test conditions. The large generic set can be leveraged as it might contain information related to the targeted specific. Its large scale allows more reliable statistical estimation and allows training higher capacity models.

Domain adaptation (Farahani et al., 2021), multi-task learning (Caruana, 1993), fine-tuning (Denevi et al., 2018) are transfer learning setups related to our setting, see Appendix A for a discussion on the relevant terminology. It is important to note that in our setting the targeted task is known at the beginning of learning (which is not a necessity for fine tuning) and that only the specific task accuracy matters (unlike multi-task learning which might also target high generic accuracy). In this work, we present our work while targeting a single specific task, with the same loss as the generic loss. Our derivations can be easily extended to the case where the specific and generic task have different loss functions. It is also simple to write derivations for the case where the specific loss is a mixture over multiple targeted specific tasks. We leave this extensions for future work.

## 4   Methods

We aim to identify the parameters $\theta$ of a model that achieves good generalization performance (held-out likelihood) over the specific distribution. For that purpose, we are given a large generic training set $\mathcal{D}_{\text{generic}}$ and small specific training set $\mathcal{D}_{\text{specific}}$. We propose to formulate the generic training problem as the minimization of the weighted loss,

$$\mathcal{L}_{\text{generic}}(\theta, \alpha) = \sum_{x \in \mathcal{D}_{\text{generic}}} w(x; \alpha) \ell(x; \theta)$$

where $w(x; \alpha)$ denotes a smaller, secondary *weighting neural network* which defines a distribution over $\mathcal{D}_{\text{generic}}$, i.e. $\forall x, w(x; \alpha) > 0$ and $\sum_{x \in \mathcal{D}_{\text{generic}}} w(x; \alpha) = 1$. We denote the solution to generic training problem as

$$\theta^*(\alpha) \in \arg \min_{\theta} \mathcal{L}_{\text{generic}}(\theta, \alpha) \tag{1}$$

Our goal is to find the parameter of the weighting network such that the loss on the *specific* training set is minimal, i.e. minimizing,

$$\mathcal{L}_{\text{specific}}(\theta^*(\alpha)) := \sum_{x' \in \mathcal{D}_{\text{specific}}} \ell(x'; \theta^*(\alpha)). \tag{2}$$

with respect to $\alpha$.

### 4.1   Data Selection as a Bilevel Optimization Problem

Our notations make clear that finding the optimal weighting network can be cast as a bilevel optimization problem: with a fixed weighting network, the optimal parameters for the main model are found by minimizing the weighted loss over the generic dataset, $\mathcal{L}_{\text{generic}}$ (Equation 1). The optimal main model parameters $\theta^*$ depends explicitly on the weighting network parameters $\alpha$; indeed, changing $\alpha$ changes the optimization problem in Equation 1 and its solution. The selection of $\alpha$ is driven by the specific set loss, $\mathcal{L}_{\text{specific}}$(Equation 2).

Equation 1 and Equation 2 form a *bilevel optimization problem* (Franceschi et al., 2018): the outer problem (Equation 2) depends implicitly on $\alpha$ through the solution to the inner problem (1). One of the strengths of such a bilevel formulation is that the weighting network must adapt to the main model: the question is to learn a weighting network such that the main model trained with that network leads to good specific performance. This has the potential to go beyond a simple model-agnostic scheme that would, for instance, build $w(x)$ based on the similarity between $x$ and the specific set. While a large body of the literature is devoted to solving bilevel problems where the inner problem (Equation 1 is convex in $\theta$ (Ghadimi & Wang, 2018; Arbel & Mairal, 2021), in our case, Equation 1 corresponds to the training problem of a neural network which is non-convex. This leads to several difficulties:

- The arg min in Equation 1 is not a single element since there are multiple minimizers. Therefore, the function $\theta^*(\alpha)$ is not properly defined.

- In order to use gradient-based methods to find the optimal $\alpha$, we have to compute the approximate Jacobian of $\theta^*(\alpha)$. This is usually done using the implicit function theorem, which only applies when the loss function in equation 1 is locally convex and such property is hard to check in practice.

Furthermore, we want a method with a computational cost similar to the standard training of the main model. In other words, we have enough budget to solve Equation 1 only once: learning $\alpha$ and $\theta$ must be carried out synchronously. This has an important consequence: the bilevel methods that we study update $\alpha$ based on the current main model state $\theta$ and not on the optimal solution $\theta^*(\alpha)$. Hence, this is a slight deviation from the bilevel formalism. This also means that the weighting network adapts to the current state of the main model and, ideally, tries to up-weight generic data that is useful *at the current state of learning*. We explore online algorithms to solve the bilevel problem when the main model is large. These algorithms alternate $\theta$ and $\alpha$ updates and leverage the asymmetry in computation cost between evaluating the large main model and the small auxiliary weighting network.

## 4.2 Updating the main model

To update the main model, we fix $\alpha$ and do a step to minimize Equation 1. A first, natural idea would be to take a mini-batch of generic data $B_{\text{generic}}$ of size $b$, compute the corresponding gradient $g = \frac{1}{b} \sum_{x \in B_{\text{generic}}} w(x; \alpha) \nabla_\theta \ell(x; \theta)$ and then use it to update $\theta$, either implementing SGD by doing $\theta \leftarrow \theta - \eta \times g$ with $\eta > 0$ a learning rate, or by using it into a more involved optimizer like Adam. However, the computation of $g$ with the previous equation can be wasteful when a significant fraction of the examples of $B_{\text{generic}}$ are assigned small weights $w(x; \alpha)$. These examples do not contribute much to $g$ while still requiring the expensive computation of their gradient $\nabla_\theta \ell(x; \theta)$.

To accelerate the optimization of $\theta$, we leverage the asymmetry between the cost of evaluating the weighting network and the main model: computing $w(x; \alpha)$ only requires inference of a small network while computing $\nabla \ell(x; \theta)$ requires inference *and* back-propagation through a large network. We start by sampling a large batch $B_{\text{generic}}^{\text{big}}$ from the generic dataset and compute $w(x; \alpha)$ for each $x$ in $B_{\text{generic}}^{\text{big}}$. From there we can take a smaller batch $B_{\text{generic}}^{\text{small}}$ from $B_{\text{generic}}^{\text{big}}$, either by sampling from the distribution defined by $w(x; \alpha)$ or by taking the examples with the highest $w(x; \alpha)$. The first option is an unbiased solution corresponding to importance sampling, while the second option is biased but observed to work better in practice. In both cases, we compute the gradient to update $\theta$ with uniform weights, using $g = \frac{1}{b} \sum_{x \in B_{\text{generic}}^{\text{small}}} \nabla_\theta \ell(x; \theta)$.

## 4.3 Updating the weighting model

With scalability in mind, we only consider *stochastic* methods, i.e., that update the weighting network parameters $\alpha$ using only a mini-batch of specific data $B_{\text{specific}}$ and a mini-batch of generic data $B_{\text{generic}}$. We consider three alternatives to update the weighting model.

Before describing alternative methods to update $\alpha$, we summarize our approach in Algorithm 1. We denote sample$(D, n)$ the set resulting from sampling $n$ times uniformly from a set $D$. We denote filter$(D, \alpha, n)$ the result from either (a) sampling $n$ times from $D$ i.i.d relying on the distribution induced by the weighting model at $\alpha$, or (b) selecting the top-$n$ highest weighted examples from $D$. The batch sizes $b_{\text{small}}, b_{\text{large}}$ are hyper-parameters selected through validation.

---

**Algorithm 1** Scalable, Online Bilevel Data Selection

**Require:** $\mathcal{D}_{\text{generic}}, \mathcal{D}_{\text{specific}}, b_{\text{small}}, b_{\text{large}}$     ▷ Training datasets, batch sizes.
    $\theta_0 \leftarrow \text{main\_model\_initializer}()$
    $\alpha_0 \leftarrow \text{weight\_model\_initializer}()$
    **for** $t = 1, \ldots, T$ **do**
       ▷ Sample generic and specific batch.
       $B_{\text{generic}} \leftarrow \text{sample}(\mathcal{D}_{\text{generic}}, b_{\text{large}})$
       $B_{\text{specific}} \leftarrow \text{sample}(\mathcal{D}_{\text{specific}}, b_{\text{small}})$

       ▷ Sample generic sub-batches.
       $B_{\text{filtered}} \leftarrow \text{filter}(B_{\text{generic}}, \alpha_{t-1}, b_{\text{small}})$
       $B'_{\text{generic}} \leftarrow \text{sample}(B_{\text{generic}}, b_{\text{small}})$

       ▷ Inner and outer updates.
       $\theta_t \leftarrow \text{update\_main\_model}(B_{\text{filtered}}, \theta_{t-1})$
       $\alpha_t \leftarrow \text{update\_weight\_model}(B'_{\text{generic}}, B_{\text{specific}}, \theta_t, \alpha_{t-1})$
    **end for**
    **return** $\theta_T$     ▷ Trained main model.

---

### 4.3.1 One gradient step unrolling - differentiable data selection (DDS)

This method is similar to (Wang et al., 2020), and updates the weighting network by doing a descent step on the loss

$$\mathcal{L}(\alpha) = \sum_{x' \in B_{\text{specific}}} \ell'(x'; u(\theta, \alpha)) \text{ with } u(\theta; \alpha) = \theta - \rho \times \sum_{x \in B_{\text{generic}}} w(x; \alpha) \nabla_\theta \ell(x; \theta), \quad (3)$$

which corresponds to the value of the specific loss on the mini-batch $B_{\text{specific}}$ after a gradient descent step for $\theta$ on the generic mini-batch $B_{\text{generic}}$ using the current weights. The idea behind this method is that $u(\theta, \alpha)$ is a reasonable approximation to $\theta^*(\alpha)$. This method requires backpropagating through a gradient descent step, which requires only a little overhead compared to a standard gradient computation. In the limit where the step size $\rho$ in the gradient update $u(\theta, \alpha)$ goes to 0, we see that $\mathcal{L}(\alpha) \simeq \rho \langle g_{\text{specific}}, g_{\text{generic}} \rangle$, with $g_{\text{specific}} = \sum_{x' \in B_{\text{specific}}} \nabla \ell'(x'; \theta)$ and $g_{\text{generic}} = \sum_{x \in B_{\text{generic}}} w(x; \alpha) \nabla \ell(x, \theta)$. Hence, the loss $\mathcal{L}$ approximately measures the alignement between specific and generic gradients. Taking derivatives gives $\nabla \mathcal{L}(\alpha) \simeq \rho \sum_{x \in B_{\text{generic}}} \langle g_{\text{specific}}, \nabla \ell(x, \theta) \rangle \nabla w(x; \alpha)$.

### 4.3.2 Stochastic Bilevel Algorithm (SOBA)

We also implement the SOBA method of (Dagréou et al., 2022), which is a scalable method to solve the bilevel problem, developed in a setting where the inner function (Equation 1) is convex. This algorithm approximates a gradient descent on $h(\alpha) = \mathcal{L}_{\text{specific}}(\theta^*(\alpha))$. The chain rule gives $\nabla h(\alpha) = \frac{\partial \theta^*}{\partial \alpha} \nabla \mathcal{L}_{\text{specific}}(\theta^*(\alpha))$. The optimum $\theta^*(\alpha)$ satisfies the first order condition $\nabla_\theta \mathcal{L}_{\text{generic}}(\theta^*(\alpha), \alpha) = 0$. Under the assumption that the Hessian $\nabla_{\theta\theta}^2 \mathcal{L}_{\text{generic}}(\theta^*(\alpha), \alpha)$ is invertible, the implicit function theorem applied to the previous equation gives $\frac{\partial \theta^*}{\partial \alpha} = -\nabla_{\alpha\theta}^2 \mathcal{L}_{\text{generic}}(\theta^*(\alpha), \alpha) \left[ \nabla_{\theta\theta}^2 \mathcal{L}_{\text{generic}}(\theta^*(\alpha), \alpha) \right]^{-1}$, which overall yields $\nabla h(\alpha) = -\nabla_{\alpha\theta}^2 \mathcal{L}_{\text{generic}}(\theta^*(\alpha), \alpha) \left[ \nabla_{\theta\theta}^2 \mathcal{L}_{\text{generic}}(\theta^*(\alpha), \alpha) \right]^{-1} \nabla \mathcal{L}_{\text{specific}}(\theta^*(\alpha))$. SOBA approximates this quantity in two ways: first, $\theta^*(\alpha)$ is replaced by the current iterate $\theta$ in the above gradient. Second, in addition to $\theta$ and $\alpha$, SOBA has an additional variable $v$ of the same size as $\theta$ that keeps track of the quantity $-\left[ \nabla_{\theta\theta}^2 \mathcal{L}_{\text{generic}}(\theta, \alpha) \right]^{-1} \nabla_\theta \mathcal{L}_{\text{specific}}(\theta)$. This is done using the stochastic iterations $v \leftarrow v - \eta \times dv$ with $dv = \sum_{x \in B_{\text{generic}}} w(x; \alpha) \nabla^2 \ell(x; \theta) v + \sum_{x' \in B_{\text{specific}}} \nabla \ell'(x'; \theta)$. The first part in $dv$ is a Hessian-vector product that can be computed efficiently at a cost similar to that of a gradient (Pearlmutter, 1994). Then, the parameters $\alpha$ are moved in the direction $d\alpha = \sum_{x \in B_{\text{generic}}} \langle \nabla \ell(x; \theta), v \rangle \nabla w(x; \alpha)$, which is a stochastic approximation of $\nabla_{\alpha\theta}^2 \mathcal{L}_{\text{generic}}(\theta, \alpha) v$, which is itself an approximation of $\nabla h(\alpha)$.

### 4.3.3 Aligned NOrmalized GRADient (Anograd)

We derive Anograd (Aligned NOrmalized GRADient) as a variant of DDS which relies on steepest descent (Boyd & Vandenberghe, 2004). We apply the steepest descent algorithm to the specific loss $\theta \to \mathcal{L}_{\text{specific}}(\theta)$. We recall that the steepest normalised descent direction according to the Euclidean norm $\|\cdot\|$ for the specific loss is

$$\Delta\theta_{\text{nsd}} = \arg\min_v \{ v^\top \nabla_\theta \mathcal{L}_{\text{specific}}(\theta) : \|v\| = 1 \} \quad (4)$$

This direction aligns with the opposite gradient when $v$ is not further constrained. In our case, $\theta$ updates should correspond to gradient descent updates on the weighted generic loss. We therefore constraints $\theta$ updates to decompose as an afine combination of individual generic example gradients, i.e. $\sum_{i=1}^{n_g} a_i \nabla \ell(x_i^g, \theta)$ where $\mathcal{D}_{\text{generic}}$ is denoted as $\{x_i^g\}_{i=1}^{n_g}$ and $a_i > 0, \forall i$. Therefore, we need to solve Equation 4 with the constraint $v \in \mathcal{V}$, with $\mathcal{V} = \left\{ \sum_{i=1}^{n_g} a_i \nabla \ell(x_i^g, \theta), \forall a_i \geq 0 \right\}$. This amounts to solving

$$\min_a \left( \frac{\sum_{i=1}^{n_g} a_i \nabla \ell(x_i^g, \theta)}{\| \sum_{i=1}^{n_g} a_i \nabla \ell(x_i^g, \theta)) \|} \right)^\top \nabla_\theta \mathcal{L}_{\text{specific}}(\theta)$$

which itself is equivalent to solve $\min_a \text{ cosine} \left( \sum_{i=1}^{n_g} a_i \nabla \ell(x_i^g, \theta), \nabla_\theta \mathcal{L}_{\text{specific}}(\theta) \right)$ We now parameterize $a$ as the output of the weighting network and introduce the loss,

$$\mathcal{L}_{\text{anograd}}(\theta, \alpha) = \text{cosine} \left( \nabla_\theta \mathcal{L}_{\text{generic}}(\theta, \alpha), \nabla_\theta \mathcal{L}_{\text{specific}}(\theta) \right).$$

The anograd method performs gradient descent on that loss to update $\alpha$. Like for DDS and Soba, we perform a single step before updating $\theta$. For scalability we rely on stochastic (batch) estimates for both both terms in the cosine. Compared to DDS, the normalization in anograd reduces the benefit of up-weighting generic examples with high gradient norm.

# 5 Experiments & Results

Our experiments focus on three application domains: language modeling, machine translation and image classification. Before introducing our experimental setup and discussing our results on each domain, we describe the baselines we considered.

## 5.1 Evaluated Alternative Methods

For our empirical comparison, we first consider two common, simple methods which do not rely on data selection. We call *baseline* pretraining on the generic training set followed by fine tuning on the specific set. We call *mixing* pretraining on a mix of generic and specific data. Each training batch contains a fixed fraction of specific data. This fraction is selected by validation.

Among data selection alternatives, we first consider *contrastive data selection*, CDS (Moore & Lewis, 2010; van der Wees et al., 2017; Wang et al., 2018). This method has four phases: (i) an initial model is pre-trained on the generic dataset, (ii) this model is fine tuned on the specific data, (iii) the generic set is restricted to the generic data whose loss improvement between the pre-trained model (i) and the fine tuned model (ii) is the greatest. Finally, (iv) the training of the pre-trained model (i) is resumed on the selected data from stage (iii). As we do for all data selection method, we consider further fine tuning the final CDS model on the specific training set. Although CDS is a generic method applicable to any training objective, it enjoys additional properties when applied to generative models trained to maximize the (conditional) training likelihood. It can both be considered an importance sampling method and an influence function based selection method (Grangier & Iter, 2022).

We also consider a *domain classifier*. In that case, a simple model is pretrained on a binary classification problem to distinguish between generic and specific training examples. The model has the same architecture as the weighting model we use with bilevel methods and it minimizes the binary cross entropy on batches with the same proportion of specific and generic data. This model can estimate the probability that an example belongs to the specific set and is applied to restrict the generic set to the data with the highest estimates. We can train a model on this restricted set and later fine tuning on the specific data.

Closer to our bilevel selection methods, we evaluate learning to re-weight, LTR (Ren et al., 2018) and meta-weight net (Shu et al., 2019)). Learning to re-weight is similar to the DDS approach we presented in Section 4 except it does not maintain a weighting model. Instead, at each step, the model considers a uniform distribution over the generic batch. It then computes the gradient of this flat weighting as free parameters with the outer update, Equation 2. This single step updates from uniform is then used to reweight the generic loss and update the main model, Equation 1. Compared to our work, this method does not persist a weighting model across steps and does not allow learning complex distributions. The lack of weighting model is also less efficient since a generic example $x$ cannot be discarded without evaluating the main model and its gradient at $x$.

Meta-weight net is a particular, simple case of DDS in which the weight model takes as input a single scalar for each example: the example loss, i.e. $w(x; \alpha) = \mathrm{mlp}(\ell(x; \theta); \alpha)$. This parameterization is sensible for some applications, e.g. loss based up-weighting is a common approach for active learning in classification problems with little intrinsic uncertainty (Settles, 2009). Loss values can be indicative of an example difficulty and loss based up-weighting might accelerate learning. However, an example loss seems to be orthogonal to the example value for domain transfer.

Table 1: Model architectures

**Language Model**

*Main model:* Transformer decoder with 12 layers, 8 attention heads, residual dimension of 256, feed-forward latent dimension of 1,024.
*Weight model:* Convolutional network with 2 layers followed by mean pooling, latent dimension of 128.

**Translation Model**

*Main model:* Transformer with 6 encoder layers and 6 decoder layers, 16 attention heads, residual dimension of 1,024, feed-forward latent dimension of 4,096.
*Weight model:* Embedding layer of dimension 32 followed by an MLP with a latent dimension of 128.

**Image Classifier**

*Main model:* Dual encoder clip model with ResNet 50 for images (224x224) and an multi-layer perceptron (2 latent layers with dim. 768) on top of sentence BERT for text.
*Weight model:* Convolutional network over 32x32 images with 4 layers of dimension 32, 32, 32 and 64.

## 5.2 Language Modeling

Our language modeling (LM) experiments relies on two datasets, the C4 dataset (Raffel et al., 2019) is used as the generic set and the RCV1 (Lewis et al., 2004) dataset is used as the specific set. C4 is a dataset of English language web pages from common crawl (Patel, 2020), while RCV1 consists of newswire stories from Reuters. This setup is representative of a generic large corpus spanning different types of examples (c4) while the specific task contains an homogeneous set of examples from the same domain and from the same source (RCV1). In our setup, we use 30m examples from C4 and 10k examples from RCV1.

Our language model is a byte-level language model based on the transformer decoder architecture (Vaswani et al., 2017). Although sub-word language models are more common than byte-level ones (Sennrich et al., 2016; Al-Rfou et al., 2019), we rely on bytes to avoid our out-of-domain generalization results to be contaminated by the mismatch between the evaluation data and the tokenizer training data (Rust et al., 2021). The weighting network is a small convolutional network. Table 1 gives architectural details. We also use the same architecture for the domain classifier baseline. We report performance in terms of log-perplexity, i.e. negative log likelihood. Our implementation is derived from the language model of the Flax library (Heek et al., 2020).

Table 2 reports the results of our language modeling experiments. In general, domain adaptation is beneficial in our setting. The only method trained exclusively on c4 (baseline without fine-tuning) is much worse than all alternatives except for MetaWeightNet. Before pretraining, mixing is the only method which directly applies updates from the specific training data and it performs best. The other methods only emphasize part of the generic set without applying specific updates during pretraining. This emphasis already show a benefit at pretraining time. More importantly, this benefit is complementary to fine tuning (Iter & Grangier, 2021) and these methods yield better results that mixing+fine-tuning. Among them, bilevel methods perform best, with SOBA giving the highest held-out specific likelihood.

We perform additional language modeling experiments with different domains. We take 9 domains from (Gao et al., 2021) and relies on 10k specific document for each domain. The generic set (c4 dataset) and the model architectures are the same as in the previous experiments. Data selection methods show that it is helpful to emphasize part of the generic set and that this emphasis is complementary to the benefit of fine tuning. The benefit varies across domains. For instance openweb is similar to the generic set c4 and only modest gains are observed, while freelaw contains legal proceedings whose domain is surely relatively rare in c4. Among methods, CDS and classifier provides a strong benefit for some datasets, but only SOBA consistently ranks among the best methods.

Table 2: Language modeling: Log-perplexity (negative log-likelihood per byte) on specific (Reuters). Circled numbers indicate the best results.

| Method | Pre-train | Fine-tune |
|---|---|---|
| Baseline | 1.197 | 0.864 |
| Mixing | 0.860 | 0.846 |
| CDS | 1.071 | 0.830 |
| Domain classif. | 1.099 | 0.892 |
| MetaWeightNet | 1.212 | 0.867 |
| LTR | 1.150 | 0.877 |
| Sparse DDS | 1.033 | ② 0.822 |
| Sparse Anograd | 1.035 | ② 0.822 |
| Sparse SOBA | 1.018 | ① 0.819 |

Table 3: Machine translation: BLEU and loss on specific (newstest2020).

| Method | Pre-train | | Fine-tune | |
|---|---|---|---|---|
| | BLEU | loss | BLEU | loss |
| Baseline | 27.63 | 2.56 | 34.06 | 2.53 |
| Mixing | 31.34 | 2.60 | 33.11 | 2.69 |
| CDS | 34.14 | 2.53 | 34.25 | 2.53 |
| Domain classif. | 35.56 | 2.37 | ① 38.03 | 2.35 |
| MetaWeightNet | 26.81 | 2.59 | 33.34 | 2.53 |
| LTR | 28.60 | 2.73 | 31.15 | 2.71 |
| Sparse DDS | 33.53 | 2.46 | 35.83 | 2.44 |
| Sparse Anograd | 36.06 | 2.41 | ② 37.28 | 2.40 |
| Sparse SOBA | 34.23 | 2.39 | ③ 37.16 | 2.38 |

Table 4: Language modeling: Log-perplexity on specific for Pile domains.

| Method | arxiv | europarl | freelaw | gutenb. | opensub. | openweb. | pmed abs | stackex. | wikipedia |
|---|---|---|---|---|---|---|---|---|---|
| Base | 1.438 | 2.219 | 1.555 | 1.365 | 1.277 | 1.220 | 1.088 | 1.313 | 1.103 |
| + ft. | 0.898 | 0.993 | 0.603 | 0.488 | 1.058 | 1.180 | 0.870 | 1.039 | 0.877 |
| Mixing | 0.909 | 1.019 | 0.606 | 0.487 | 1.067 | 1.153 | 0.874 | 1.049 | 0.874 |
| + ft. | 0.899 | 1.081 | 0.600 | 0.478 | 1.059 | 1.156 | 0.860 | 1.042 | 0.865 |
| CDS | 1.216 | 1.981 | 1.284 | 1.253 | 1.193 | 1.154 | 0.829 | 1.100 | 0.944 |
| + ft. | ① 0.861 | 0.977 | 0.614 | 0.482 | ② 1.039 | ③ 1.131 | 0.791 | ① 0.971 | ③ 0.823 |
| Classifier | 1.293 | 1.608 | 1.133 | 1.202 | 1.266 | 1.139 | 0.787 | 1.159 | 0.914 |
| + ft. | 0.920 | ② 0.892 | ③ 0.582 | 0.481 | 1.066 | ① 1.125 | ① 0.765 | 0.998 | ② 0.807 |
| S. DDS | 1.231 | 1.868 | 1.184 | 1.285 | 1.288 | 1.290 | 0.828 | 1.112 | 0.988 |
| + ft. | ② 0.867 | 0.948 | 0.580 | ① 0.477 | 1.104 | 1.262 | 0.792 | ② 0.977 | 0.839 |
| S.Anograd | 1.219 | 1.659 | 1.237 | 1.274 | 1.193 | 1.150 | 0.814 | 1.132 | 0.989 |
| + ft. | ③ 0.871 | ③ 0.895 | ② 0.580 | ① 0.477 | ③ 1.042 | 1.133 | ③ 0.784 | 0.988 | 0.838 |
| S. SOBA | 1.210 | 1.582 | 1.124 | 1.296 | 1.184 | 1.149 | 0.803 | 1.134 | 0.908 |
| + ft. | 0.872 | ① 0.883 | ① 0.579 | ② 0.480 | ① 1.035 | ② 1.128 | ② 0.779 | ③ 0.989 | ① 0.803 |

## 5.3 Machine Translation

Our machine translation (MT) experiments learn a translation model from English into German. They rely on two datasets: our generic set is the Paracrawl dataset (Release 1 for WMT 2018) with 36m sentence pairs (Bañón et al., 2020). Our specific set concatenates the WMT newstest sets (2009–2019) with source original English sentences, which amounts to 10,015 sentence pairs (Akhbardeh et al., 2021). We use the 2020 newstest data (1,997 sentences) as our validation set and leave the 2021 newstest data (1,418 sentences) as our test set. Our generic set is therefore a large crawled set with different types of text and varying translation quality while the specific set is a small set from a single domain with high quality translation.

Our translation system is a sub-word model based on the transformer encoder-decoder architecture. For the weighting network and the domain classifier we compose a shared embedding layer for source and target and apply a multi-layered perceptron on the contatenated averaged embeddings of the source and target sentences. Table 1 gives architectural details. Our evaluation relies on BLEU scores (Papineni et al., 2002) for beam-4 decodes. We also reports some results in terms of loss (i.e. negative log likelihood with label smoothing strength of 0.1). Our implementation is derived from the translation model of the Flax library (Heek et al., 2020).

Table 3 reports the results of our machine translation experiments. In that case, SOBA and Anograd provide a strong improvement over the baseline, both before (more than +7 BLEU) and after fine tuning (more than +3 BLEU). However in this setting, the domain classifier is even more effective. Both for language modeling and for machine translation, we remark that MetaWeightNet performs poorly. MetaWeightNet predicts the selection weight from the loss on the example. This is a common strategy in active learning for classification (Settles, 2009). This notably assumes that the loss per example is indicative of how well the model perform on them. However, in the case of language modeling and translation, the loss per example also reflects the intrinsic entropy of the examples which varies greatly across examples. It therefore seems difficult to use the loss the sole feature for data selection for these tasks. Comparing the results of LTR and DDS is also interesting as the methods are similar. DDS maintains a weighting model across steps, while LTR just adapt the distribution for each batch and does not persist cross-steps selection parameters. The benefit of DDS tells that the stability across steps and the lesser dependency on the batch size are important.

## 5.4 Image Classification

Our vision setup performs contrastive training over image and captions – CLIP (Radford et al., 2021) – for generic training and image classification for specific training. Specifically, contrastive learning should select the correct caption within a large set of random captions. This approach also allows to perform classification by representing classes as captions of the form "a photo of a <class name>" and letting the model infer the most appropriate caption within that set. As datasets, we rely on yfcc15m (Radford et al., 2021) for generic training (14.9m image/caption pairs) and ImageNet67 (Eshed, 2020) dataset for the specific task. Imagenet 67 consists in 67 high level classes over Imagenet (Deng et al., 2009), e.g. building, person, fish, dog... Like for other experiments, we consider a setup with limited specific data and take 2,010 specific examples, 30 per class, for training. Held-out evaluation is performed with 50 images per class.

For our CLIP model, the image branch is a Resnet 50 (He et al., 2016) while the text branch applies an MLP over precomputed sentence embeddings from Sentence BERT (Reimers & Gurevych, 2019). Training applies contrastive learning only over alternative captions: for the generic loss, we consider a buffer of past captions as negatives; for the specific loss, we consider all the other class captions as negatives. Our weighting network is a small convolutional network over low resolution images (32x32). Table 1 gives architectural details.

Table 5 reports the results of our image classification experiments. Unlike for our text experiments, the benefit of data selection is limited for this task. After fine-tuning, only the CDS and domain classifier methods outperform the baseline model. The bilevel data selection methods do not outperform the baseline method. We shed light on the cause of this poor performance in our further analysis in Section 6.3.

Table 5: Image Classification: Accuracy on specific (ImageNet67).

| Method | Pre-train | | Fine-tune | |
|---|---|---|---|---|
| | Acc. | Loss | Acc. | Loss |
| Baseline | 41.1 | 2.694 | 54.9 | 1.902 |
| Mixing | 55.1 | 1.928 | ③ 55.1 | 1.928 |
| CDS | 42.3 | 2.512 | ② 55.2 | 1.957 |
| Domain classif. | 44.1 | 2.571 | ① 57.5 | 1.949 |
| MetaWeightNet | 35.5 | 2.743 | 43.9 | 2.351 |
| LTR | 36.4 | 2.712 | 44.9 | 2.364 |
| Sparse DDS | 40.5 | 2.609 | 53.2 | 2.067 |
| Sparse Anograd | 41.4 | 2.563 | 53.6 | 2.055 |
| Sparse SOBA | 41.1 | 2.622 | 53.9 | 2.057 |

Table 6: Does the weight model's trajectory correspond to a curriculum? Pre-train LM log perplexity on Reuters for different trajectories.

| Weight model trajectory | log-perplexity |
|---|---|
| SOBA curriculum | 1.047 |
| Final weighting | 1.044 |
| Shuffled weighting | 1.055 |

Table 7: Alternative sampling strategies to filter the generic batch. Pre-train LM log perplexity on Reuters.

| Sampling strategy | log-perplexity |
|---|---|
| Importance Sampling | 2.038 |
| Sampling without replacement | 1.040 |
| Selecting the highest weights | 1.227 |

## 6 Analysis

### 6.1 Learning a Distribution vs Learning a Curriculum

Algorithm 1 produces a sequence of main model's parameters $\theta_t$ and weighting model's parameters $\alpha_t$, that go towards the solution of the bilevel problem (2). We investigate whether the weighting model's parameters correspond to a *curriculum*: does the evolution of the weighting parameters $\alpha_t$ adapt to the particular data needed at each step, helping the model perform better than a fixed weighting scheme?

We are in the LM task setup described in Section 5.2, except that we use a smaller "large batch size" $B_{\text{generic}}^{\text{big}}$ (see Section 4.2). We run Algorithm 1 with SOBA to obtain a sequence $\theta_t, \alpha_t$. We then compare this setting with two new training runs with standard ERM using different data weighting:
- *Final weighting:* a new main model is trained with fixed weighting from the weighting model $\alpha_T$.
- *Shuffled weighting:* a new main model is trained with a random permutation $\sigma$ of the weights $\alpha_{\sigma(t)}$.
Table 6 shows that SOBA's curriculum is not beneficial compared to the fixed final weighting scheme on this task. The lesser performance of shuffled weighting certainly highlight poor weighting from early $\alpha_t$. Note that the results reported in this section do not match Section 5.2 because of a smaller $B_{\text{generic}}^{\text{big}}$ was used in this ablation.

### 6.2 Big Batches: Importance Sampling vs Filtering

In Algorithm 1, we denote $\text{filter}(B, \alpha, n)$ the operation resulting in a smaller sub-batch of size $n$ starting from the generic batch $B$ using the weighting network parameterized by $\alpha$. To get an unbiased estimate of the re-weighted generic loss, one can apply *importance sampling* and sample (with replacement) from the weight distribution induced by $\alpha$ on B. Alternatively one can instead sample *without replacement* from that distribution or restrict the batch B to its *highest weighted* elements. The last two alternative are biased. Nevertheless, our results in Section 5 uses sampling without replacement.

Table 7 justifies this choice. Basically, we observe that the learned weighted distribution is concentrated along few examples which yield importance sampling batches to contain less diverse sets than when sampling with replacement. Similarly, cutting the tail of the distribution (highest weights selection) drop lower weighted – but still helpful – examples. These experiments illustrate that gradient-based estimates fail to account for the long term benefit of a more diverse training set. Although sampling with replacement alleviates this issue, more principled solutions should be investigated in future work.

### 6.3 On the Discriminative Power of Gradient Aligments

Our experiments highlight that bilevel optimization for data selection performs differently across tasks. We explore if a simple diagnostic could help understand these differences. Our method considers a base model $\theta_t$ trained on the generic distribution for $t$ steps. We take a diagnostic batch $B_{\text{mix}}$ which blends unseen generic and specific data in equal proportion. We want to verify how the weighting model on the mixed data would move away from a uniform weighting scheme in an outer update. We want to observe whether the weighting model would increase the weights of specific examples if some of these were "hidden" within the generic set.

For DDS and Anograd, increase or decrease in weights depends on the alignment between individual example gradients from $x \in B_{\text{mix}}$ and the expected gradient on the training specific batch $B_{\text{specific}}$.

$$a(x, B_{\text{specific}}) = \nabla_\theta \ell(x, \theta)^\top \, \nabla_\theta \ell(B_{\text{specific}}, \theta)$$

between individual example gradients from $x \in B_{\text{mix}}$ and the expected gradient on the training specific batch $B_{\text{specific}}$, denoted as $\ell(B_{\text{specific}}, \theta) := \frac{1}{|B_{\text{specific}}|} \sum_{x \in B_{\text{specific}}} \ell(x, \theta)$. We then normalize the batch gradient and define,

$$a_{\text{norm}}(x, B_{\text{specific}}) = \nabla_\theta \ell(x, \theta)^\top \, \frac{\nabla_\theta \ell(B_{\text{specific}}, \theta)}{\|\nabla_\theta \ell(B_{\text{specific}}, \theta)\|}.$$

This normalization allows to take an example $x$ and verify whether its gradient aligns better with the specific batch gradient than with the generic batch gradient, i.e.

$$a_{\text{norm}}(x, B_{\text{specific}}) > a_{\text{norm}}(x, B_{\text{generic}}).$$

We report the rate at which this inequality is true for specific examples,

$$\text{SAR} = \mathop{\mathbb{E}}_{\substack{x \sim \mathcal{D}_{\text{specific}} \\ B_{\text{specific}} \sim \text{Batch}(\mathcal{D}_{\text{specific}}) \\ B_{\text{generic}} \sim \text{Batch}(\mathcal{D}_{\text{generic}})}} \mathbb{1}\left\{a_{\text{norm}}(x, B_{\text{specific}}) > a_{\text{norm}}(x, B_{\text{generic}})\right\}$$

We call this measure the Specific Acceleration Rate, SAR. We would like this rate to be high, meaning that, according to the Taylor approximation of the loss, updates collected from a batch of specific examples should improve the loss on a given specific examples faster than updates collected from a generic batch. Symmetrically, we define the Generic Acceleration Rate,

$$\text{GAR} = \mathop{\mathbb{E}}_{\substack{x \sim \mathcal{D}_{\text{generic}} \\ B_{\text{specific}} \sim \text{Batch}(\mathcal{D}_{\text{specific}}) \\ B_{\text{generic}} \sim \text{Batch}(\mathcal{D}_{\text{generic}})}} \mathbb{1}\left\{a_{\text{norm}}(x, B_{\text{generic}}) > a_{\text{norm}}(x, B_{\text{specific}})\right\}$$

It is also desirable that this rate is high. When SAR, GAR are close to chance (50%), there are two possible explanations, (i) either generic and specific batches have the same effect on the model, meaning that data selection is unlikely to be helpful since training on generic is already as good as training on specific for the purpose of minimizing the specific loss; (ii) alternatively, the linear approximation (order 1 Taylor expansion) does not help discriminating between the effect of generic and specific examples on the specific loss. In that later case, such a learning problem will be a challenge for bilevel optimization methods where gradient alignments indicates which part of the dataset to upweight.

Table 8 reports SAR and GAR for our results. These results are indicative of the empirical benefit of bilevel optimization methods for data selection. Language modeling where DDS, Anograd and SOBA are advantageous, has the highest SAR. Conversely, our image classification problem shows near random SAR, GAR in line with the poor performance on bilevel methods on this problem. We therefore consider that measuring SAR/GAR can be a simple but informative diagnostic to assess the potential benefit of bilevel methods on a new problem.

Table 8: Specific & Generic Acceleration Rates

| Task | | SAR | GAR |
|---|---|---|---|
| Language modeling | (c4, reuters) | 86.2% | 69.4% |
| Machine translation | (paracrawl, newstest) | 78.1% | 68.2% |
| Image classification | (yfcc15m, imagenet67) | 50.3% | 49.8% |

Table 9: Small and base model architectures for scaling up the language modeling task.

| Model | Layers | Res. dim. | ff dim. | # Params |
|---|---|---|---|---|
| Small | 4 | 128 | 512 | 824.064 |
| Large | 12 | 256 | 1024 | 9.530.880 |

Table 10: Results of the scaling experiment

| Model | Method | Pre-train |
|---|---|---|
| Small | Sparse SOBA | 1.292 |
| Large | Baseline | 1.197 |
| Large | Sparse SOBA | 1.018 |
| Large | Weights from Small | 1.034 |

### 6.4 Re-using Weighting Strategies with Larger Scale Models

The weighting network is trained by solving the bilevel problem (2), where the loss function $\ell$ depends on the model's architecture. We investigate whether a weighting network learned with a small model can be re-used out-of-the-box to train a large model and get good performances on the specific set. The weighting network is frozen: the large model is trained by solving $\min_\theta \sum_{x \in \mathcal{D}_{\text{generic}}} w(x; \alpha)\ell(x; \theta)$ with $\alpha$ fixed to the final parameters of the weighting network trained with the small model. We perform this experiment on the language modeling task (Section 5.2), where the small model and large model architectures are specified in Table 9; the large model's architecture is the same as in Section 5.2, and it has about ten times more parameters. We observe that the weighting network learned with the small model transfers to the large architecture and leads to a large decrease in the loss on the specific set, which is only slightly worse than using the Sparse SOBA method on the large model itself. This means that the weighting network learned at a small scale can then seamlessly be used at a larger scale and lead to significant performance improvement on the specific set.

## 7 Conclusions

This work studies bilevel optimization for learning training distributions. We consider the setup where a model is trained from two training sets, a large generic set and a small specific set, where only the later is representative of the test conditions. We propose a scalable algorithm that learns a training distribution over the generic data such that the loss on the specific set is minimized. We showed that our formulation gathers independently-proposed gradient-based methods for data selection under a common framework. We introduced an algorithm that enables streaming through the generic dataset quickly by examining most of the generic samples with only an inexpensive small auxiliary model. This work reported a comprehensive and realistic empirical comparison of data selection strategies across language modeling, machine translation and computer vision. We studied the conditions in which gradient-based data selection is effective and propose a diagnostic based on gradient alignment to efficiently assess these conditions.

Our work also delineates interesting questions for future work. Conceptually, we observe that gradient-based selection methods fail to reward properly the diversity of the selected samples (Section 6.2), which deserves further theoretical study. Empirically, the complementarity between fine-tuning and generic data selection highlights that the updates collected from re-weighted generic training and from specific training are different. The existence of complementary updates and their exploitation might also be possible even when one is presented with a single monolithic training distribution.

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

## A Common Settings in Transfer Learning

The transfer learning literature defines various settings to leverage training data from a different task and/or distribution. Although not all papers use the same definitions, Table 11 presents the most common settings with reference to the literature supporting these definitions.

Table 11: Classical Transfer Learning Settings

---

**Transfer Learning** leverages a source distribution in order to perform better on a target distribution (Thrun & Pratt, 1998).

**Multitask Learning** improves generalization by leveraging the information contained in the training signals of related tasks (Caruana, 1993; 1997).

**Domain Adaptation** aims to improve accuracy on target distribution with insufficient labeled data by leveraging a model trained on a different but related source distribution (Farahani et al., 2021).

**Unsupervised Domain Adaptation** considers the setting where labeled source domain data (x, y) are available for training, while only unlabeled (x) data from the target domain are available (Ganin & Lempitsky, 2015).

**Distribution Shift** considers that the test distribution is different from the training distribution, usually in the context where the model cannot be retrained to adapt to the new test conditions (Koh et al., 2021).

**Gradual Distribution Shift** is an online setting where the training distribution progressively evolves (Kumar et al., 2020).

**Covariate shift** corresponds to a predictive setting where the distribution over the input features $p(x)$ is different at training and test time, while the posterior distribution $p(y|x)$ does not change (Bickel et al., 2009).

**Label shift** corresponds to a predictive setting where the class prior $p(y)$ between train and test changes but the conditional distribution $p(y|x)$ is assumed identical (Garg et al., 2020).

**Fine-tuning** is a specific domain adaptation technique which considers training a model on the target domain from an initial model trained on the source domain (Matic et al., 1993; Denevi et al., 2018; Zhang et al., 2021).

**Zero-Shot Task Transfer** addresses new tasks at test time without updating the model (Larochelle et al., 2008). It typically relies on a way to represent novel tasks in order to condition the model, e.g. text prompts (Radford et al., 2019; Srivastava et al., 2022).

**Few-Shot Task Transfer** is similar to zero-shot transfer and does not update the model weights. As a difference, the task conditioning information provides few training instances along with the description of the tasks (Radford et al., 2019; Srivastava et al., 2022).

---

