# OpenReview forum: "Adaptive Training Distributions with Scalable Online Bilevel Optimization"
_TMLR — Rejected by TMLR_

### Review · Reviewer_LxYe · 2023-12-22

**Summary Of Contributions:**

This paper follows the previous formulation of selecting the most relevant samples to the task-specific data from a large generic pretraining dataset as a bilevel optimization problem.

The inner problem aims to learn a prediction model $\theta$ from the pretraining data given the weighting neural network param $\alpha$, and the outer problem is to optimize $\alpha$.

Three solutions have been considered to solve the above problem: 1. differentiable data selection (DDS), which conducts direct gradient updates. 2. Stochastic Bilevel Algorithm (SOBA), which is derived from the implicit function theorem and the stochastic approximation of Hessian-vector product. 3. Anograd, which conducts gradient alignment between the pretraining data and the task-specific data.

**Audience:**

Yes

**Claims And Evidence:**

No

**Requested Changes:**

Please fix the points mentioned in weaknesses.

**Strengths And Weaknesses:**

### Strengths
1. The proposed algorithms are correct and intuitively make sense.
2. The experiments are conducted on diverse datasets for language model, translation model and image classification.
3. The analysis in section 6 is quite interesting and informative.

### Weaknesses

1. The idea of sample weighting is not that novel and lacks detailed discussion w.r.t related works in domain adaptation, e.g., [1].
2. The gradient alignment idea is quite similar to that in [2], which lacks a discussion.
3. The experimental results are reported without variance or standard deviation.
4. No training details or code or hyper-parameter settings were provided for the reproducibility of the results.
5. The writing could be further improved, and some concepts need to be briefly introduced before being used.

[1] Huang, Jiayuan, Arthur Gretton, Karsten Borgwardt, Bernhard Schölkopf, and Alex J Smola. “Correcting sample selection bias by unlabeled data”. In: Advances in neural information processing systems. 2007, pp. 601–608 (cit. on pp. 20, 34, 66).

[2] Matthew Riemer, Ignacio Cases, Robert Ajemian, Miao Liu, Irina Rish, Yuhai Tu, and Gerald Tesauro. Learning to learn without forgetting by maximizing transfer and minimizing interference. In International Conference on Learning Representations, 2019.

---

### Review · Reviewer_sdVz · 2024-01-20

**Summary Of Contributions:**

The paper proposes a adaptive training method to make the pretrained model quickly adapted into specific domains. Specifically, it jointly train a weighting network with the original model to quickly assign different weight to the training examples. It formalize the data selection problem into a bilevel optimization where the main model update's data weight could be got with the weight network's inference. For the weight network update, the paper uses three strategies: DDS, SOBA and anograd. DDS only do one time unrolling to achieve a update, while SOBA utilizes the second order information by assuming the inner loop as a convex problem. Anograd also unroll the update once but with a different loss function. The experiments are conducted in three settings including language model, machine translation and image classification from CLIP. The results show the proposed method could achieve a slightly better performance on NLP task but worse performance in image classification.

**Audience:**

Yes

**Broader Impact Concerns:**

I don't see any ethic concerns.

**Claims And Evidence:**

Yes

**Requested Changes:**

1. The definition of $\alpha$ and $x'$ should be explained just after eq 1 and eq2. They are introduced later but I prefer to see it at the first place they appeared.
2. Section 5.1 should be compressed in a better way. Some of the description should be in related works rather than in the experiments.
3. Missing reference with Table 4. Also, Table 3 is a section 5.3's results that appears before the section 5.2's results Table 4. It will cause unnecessary confusion. Those 3 tables could be organized in a better way to save some space.
4. Consider to add some discussion on how to apply it into mutli-task scenario to improve paper's novelty.

**Strengths And Weaknesses:**

Pros:
1. The paper is well-written and easy to follow. There are enough information for someone not working in this area to know the proposed method and related work. However, some parts of papers could be further improved.
2. The ablation study on analyzing why the proposed framework fails and works is inspiring and interesting. I appreciate the paper's efforts to admit its weaknesses on image classification and try to dig out what the reasons are behind.

Cons:
1. I feel the setting is a little impractical where only a single down-stream task is already known at the beginning of learning. I fail to see how the proposed framework could easily adapted into multi-task version.
2. The paper's novelty is not significant. Also pointed out by the paper, the proposed framework shares a lot of similarity with previous method and the improvement is mainly with introducing a weighting network to do the data selection task.
3. The proposed method's performance is not significant. The NLP tasks does show some improvement but pretty marginal in my opinion. And the proposed method gets worse performance in the image domains. Although the paper has proposed the potential reason, the results questioned the proposed method could be used into real-application where data and applications are more diverse.

---

### Review · Reviewer_tbsW · 2024-01-29

**Summary Of Contributions:**

This paper studies the distribution shift problem (as claimed) that arises when applying large-scale models pre-trained on web-scale corpora. The paper claims that these models struggle with generalizing to test applications with different distributions than the training data, and proposes a scalable data selection strategy to efficiently track the direction of updates that will make the model quickly adapt to target distribution. Extensive experimental results on text and multimodal image-text modeling tasks show that the proposed method sometimes outperforms comparable baselines.

**Audience:**

No

**Claims And Evidence:**

No

**Requested Changes:**

N/A

**Strengths And Weaknesses:**

Strength:

1. The paper is well written and easy to follow
2. The paper unifies different gradient-based data selection methods into one analytical framework.
3. Experimental results consider both NLP and Vision (Multimodal modeling) tasks, which is fairly comprehensive among data selection / distillation literature.

Weakness:

1. My main concern is about the motivation of this work. The central problem this paper addresses, as stated by the authors, is the distribution shift (generalization issue) of large models pretrained on web-scale data. However, nowadays LLMs and VLMs pretrained on web-scale data show strong generalization ability, demonstrated by their ability to zero-shot transfer to a wide-range of tasks with different test distributions. It is unclear whether the problem this paper solves is relevant.
2. The scale of experiments (model and dataset size) is very limited, and it is unclear how well the proposed method actually generalize to real-world scenarios.
3. The performance of the proposed method is not consistently better than existing ones.

---

### Decision · Action_Editor_h7eH · 2024-03-17

**Recommendation:** Reject

**Comment:**

This paper studies the distribution shift problem in large-scale models pre-trained on web-scale corpora. The paper claims that these models struggle to generalize to test applications under test distribution shift, and proposes a scalable data selection strategy to efficiently track the direction of updates that will make the model quickly adapt to the target distribution.

All reviewers agreed that this is an interesting problem and that the paper is clearly written. However, several concerns were raised about the paper's supporting evidence:
- Motivation. Reviewer tbsW pointed out that LLMs might have the ability to transfer zero-shot to a wide range of tasks with different test distributions. It is unclear whether the problem this paper solves is relevant.
- Several reviewers felt that the problem setting was a bit impractical where only a single downstream task is already known. They would suggest experiments in real-world scenarios.
- More importantly, the authors still did not provide a rebuttal and revision of the paper to address reviewers’ concerns.

**Decision**: In the recommendation, all reviewers felt that the current version was still insufficient for acceptance. This AE supported this decision and encouraged the authors to include these suggestions in their future submissions.

**Audience:**

Yes.

**Claims And Evidence:**

No. Please see the comment.

**Resubmission Of Major Revision:**

The authors may consider submitting a major revision at a later time.